# Understanding Online Voluntary Self-Exclusion in Gambling: An Empirical Study Using Account-Based Behavioral Tracking Data

**DOI:** 10.3390/ijerph18042000

**Published:** 2021-02-19

**Authors:** Maris Catania, Mark D. Griffiths

**Affiliations:** 1Kindred Group, Tigne Point, Sliema TPO0001, Malta; 2Psychology Department, Nottingham Trent University, 50 Shakespeare Street, Nottingham NG1 4FQ, UK; mark.griffiths@ntu.ac.uk

**Keywords:** gambling operators, responsible gambling, voluntary self-exclusion, behavioral tracking, consumer protection

## Abstract

Online gambling has continued to grow alongside new ways to analyze data using behavioral tracking as a way to enhance consumer protection. A number of studies have analyzed consumers that have used voluntary self-exclusion (VSE) as a proxy measure for problem gambling. However, some scholars have argued that this is a poor proxy for problem gambling. Therefore, the present study examined this issue by analyzing customers (from the gambling operator *Unibet*) that have engaged in VSE. The participants comprised of costumers that chose to use the six-month VSE option (n = 7732), and customers that chose to close their *Unibet* account due to a specific self-reported gambling addiction (n = 141). Almost one-fifth of the customers that used six-month VSE only had gambling activity for less than 24 h (19.15%). Moreover, half of the customers had less than seven days of account registration prior to six-month VSE (50.39%). Customers who use VSE are too different to be treated as a homogenous group and therefore VSE is not a reliable proxy measure for problem gambling. The findings of this research are beneficial for operators, researchers, and policymakers because it provides insight into gambling behavior by analyzing real player behavior using tracking technologies, which is objective and unbiased.

## 1. Introduction

Through the internet, and especially via smartphones, gambling has changed into an activity that can be done anywhere at any time [1]. In fact, in the 2010 British Gambling Prevalence Survey, it was reported that up to 14% of the total adult population had gambled online [2] and had increased to 21% in the latest British study [3]. This form of gambling has increased in popularity but can bring about risks for problem gamblers, due to the possibility of offering mood-altering experiences such as immersion and escapism, which can in turn lead to disproportionate involvement [4]. Online gambling has expanded quickly but gambling regulation to protect players and minimize harm has tended to lag behind. Due to the global nature of online gambling, it has become accessible across national borders. While gambling regulations were traditionally for territory-based gambling, many countries have not got to grips with online regulation of the activity [5]. Subsequently, there is a high number of gambling operators based in Malta, in which the local authority imposes license requirements for consumers in other countries, making borderless gambling possible [6].

It is understandable that the more popular online gambling becomes, the more concerning it may be because it might lead to an increase in gambling-related harm [7]. Internet gambling is often seen negatively due to its high accessibility and convenience [8,9], and it may increase the occurrence of problem gambling due to higher gambling exposure [10,11]. However, due to factors concerning innovative new technology, better consumer protection may be achieved [12,13]. Moreover, online gambling operators may provide better and more accessible responsible gambling tools than those provided offline [14,15,16]. 

Responsible gambling and consumer protection have been discussed and defined for several years [17]. Criticism lies in the terminology of the words ‘responsible gambling’ because the shift of responsibility is placed onto the consumer, where the consumer needs to utilize the responsible gambling tools offered by the operator [18]. Further arguments have also been raised because there may be a conflict of interest for the operator to invest in consumer protection since this might have an impact on the commercial interest and financial gain of the operator [19]. Despite this, it can be evident that current legislation may impose a financial strain on the operator if responsible gambling interventions are not carried out properly and may result in large fines for the operators by regulatory bodies [20]. Gambling may result in harm that is experienced by different social groups such as families and communities. Consequently, harm is not only experienced by individuals suffering from a gambling disorder [21]. Harms that are common among individuals with gambling disorder include escalating levels of gambling that go beyond disposable income, relationship problems with family and loved ones, health problems, and compromising occupation and/or education [22,23]. 

Online gambling has provided new opportunities for data to be analyzed for consumer protection purposes [24,25]. This is possible due to the fact that all gambling transactions records are saved and stored for each consumer [26]. Through its online nature, online gambling operators may offer different tools such as deposit limits, play limits and/or loss limits, which can either be imposed or suggested to the consumers [27]. Research conducted by Wood, Shorter and Griffiths [28] concluded that voluntary self-exclusion (VSE) was one of the highly recommended tools that gambling operators should utilize. VSE is a responsible gambling tool that removes an individual’s access to gambling with gaming operators [29,30]. In an early study by Smeaton and Griffiths [31], 30 major gambling operators were evaluated, and it was reported that only one operator at that time had this option available. Due to increased regulatory pressures and an increase in importance for consumer protection measure, this has changed markedly. For instance, in a study conducted by Bonello and Griffiths [32], 50 major gambling operators were evaluated, and VSE was offered by 86% of these operators, a large increase on the study by Smeaton and Griffiths [31] in terms of VSE availability. The online gambling industry now has the possibility of monitoring and saving customers’ activity data at minimal costs for the operator [20,33,34]. Using these tracking data, responsible gambling tools may support operators in giving personalized tips and communication to consumers to regulate their gambling [1]. Data from online behavioral tracking can be used to assess gambling intensity by players. An initial simulation study by Auer, Schneeberger and Griffiths [35] of 300,000 gamblers developed ‘theoretical loss’, a metric that can be used to calculate gambling intensity and comprises of the amount of money wagered, multiplied by the probability of winning on the particular type of gambling activity. This metric was then tested on a real customer sample of 100,000 online gamblers using their tracking data. The findings indicated that the theoretical loss metric was robust [36] and has since been used to evaluate responsible gambling tools’ efficiency such as limit-setting [37].

In another study, customer communication was analyzed to determine whether specific indicators can be used to predict problem gambling [16]. Here, 1008 emails were analyzed from a group of customers who used VSE and a control group. It was evident from the findings that the frequency of customer service communication, and the tonality of the written correspondence may be used as a predictor of VSE. An analysis of anonymous player data provided by *GTECH G2* (an internet gambling software provider) found that customers using VSE had greater losses when compared to the control group which did not [29]. This study has its benefits because it showed gambling customer activity across different operators, but in turn this may also be a limitation because most operators will have more than one gambling software provider. *GTECH G2* offers gambling software for gambling operators, and therefore this study has its benefits because it showed gambling customer activity across different operators. Nonetheless, one gambling operator may have multiple gambling software providers, and therefore customer activity can still be limited.

Studies examining behavioral tracking were carried out extensively for a few years due to a collaboration with *Bwin Interactive*. An anonymized data sample of heavily involved gamblers was analyzed by [11]. Although this study possibly shed some insight into problem gambling by examining heavily involved gamblers, this approach may be limited because not all heavily involved gamblers will be problem gamblers. Problem gambling is also dependent on other social and economic factors. Another study from the same dataset examined the anonymized tracking data of the first 90 days of a customer’s journey with the operator [38]. The results showed that the highest betting activity was at the beginning of the player’s journey and that there was an episodic increase in betting activity every seven days. Gamblers may have an initial betting activity which is high because they may be testing the website and trying new products and features available. Another reason may be that there was an acquisition bonus that may have had to be used within a specific time period after registration. It is likely that the episodic increase in betting activity every seven days is due to the availability of sports events because most of them are played at the weekend and therefore betting tends to be episodic based on the availability of sporting events that can be bet upon. 

In another study, 2696 gamblers were evaluated by exploring their payment transactions prior to self-excluding [39]. Haeusler proposed that a potential indicator for problem gambling was an inconsistency of the amount withdrawn, which fluctuated from very high to very low amounts. Although using payment data to build indicators for online problem gambling may appear to be a good option, using VSE as a proxy measure for problem gambling might not be. This is due to the fact that not all customers choose to use VSE because of a problem with their gambling [29,40]. For example, in the aforementioned study conducted by Haeusler, it was reported that 23.3% of customers that used VSE in January 2015 had no deposit payments at all in the year before. This lack of gambling activity may be the result of the gambler being annoyed with the gambling operator and consequently wanting to close their account as a sign of their unhappiness. 

In a study by Gray et al. [7], responsible gambling interventions were examined to identify possible problem gambling indicators. These interventions included instances where players requested to change their money limits, cancelling their withdrawals, and fair play complaints, amongst others. This study showed that the players that initiated such interventions had significantly more gambling activity across different products [7]. A limitation in this sample was that it only included players who contacted customer services. Only including players that have initiated these types of interventions excludes players that might have had a responsible gambling issue, but were too ashamed, or did not want to contact customer services. A better approach was utilized by Xuan and Shaffer [41] where they analyzed an anonymized sample from *Bwin Interactive* of players that closed their gambling account online [42]. Consistent with LaBrie and Shaffer [42], the study concluded that prior to closing their account, players had increased losses and heightened risk-taking.

VSE has been used as a proxy measure for problem gambling in a number of studies [16,42,43,44]. Although it is very convenient and used in several studies, using VSE as a proxy measure for problem gambling may not always be the best approach, especially when accounting for online VSE [40]. Although there are several studies that have used VSE as a proxy measure of online problem gambling, little published empirical research has been carried out on its effectiveness. In a study by Dragicevic et al. [29], it was reported that a quarter of all the players that had self-excluded had done this on the same day that they opened the online gambling account. It is also unlikely that online VSE has the same stigma that may be present when self-excluding in a land-based venue, and that gamblers may have ulterior motives for choosing VSE.

Through online behavioral tracking, operators can tailor harm-minimization interventions [39], which is possible through the objective analysis of large sample sizes [36]. As aforementioned, VSE has been used as a proxy measure for online problem gambling in several studies. Nonetheless, gamblers who use VSE typically comprise gamblers on one part of the gambling spectrum [39]. Moreover, VSE may be used by a gambler as a responsible gambling measure, rather than a measure indicating problem gambling [10]. A study by Hayer and Meyer [45] found that 26.3% of the gamblers that used VSE had chosen to self-exclude because they were annoyed with the gambling operator. 

Providing tailored help and attempting early detection of problem gambling will help players to regulate their gambling. This will lead to more sustainable long-term revenue for the gambling operator [44]. By adopting early detection of problem gambling, several benefits may be achieved for the gambler and the gambling operator, such as minimizing (i) financial harm for the gambler, (ii) potential negative impact on the gamblers’ families, and (iii) negative psychosocial impact in the communities where problem gamblers live.

The argument that VSE is not an ideal proxy measure for problem gambling is not to be interpreted that VSE should not be offered, but this is an opportunity to better understand the potential misuse of VSE. The aim of the present paper is to evaluate whether VSE is a good proxy measure of problem gambling by examining an anonymized sample of customers that used VSE. The rationale for using gambling expenditure in this study was due to the fact that financial harm is a major issue when it comes to reporting gambling harm. Since this is an initial study looking into this area, the authors have chosen to specifically analyze this variable in relation to VSE. 

## 2. Materials and Methods

### 2.1. Participants and Procedure

The participants in the present study were all UK customers who chose to use voluntary self-exclusion or close their account for self-reported gambling addiction with *Unibet*. These participants were costumers that chose to use the six-month VSE option (n = 7732), and those that chose to close their *Unibet* account due to a self-reported gambling addiction (n = 141). All data were from January 2017 to May 2018. This group of customers comprised 80.9% males (n = 6369) and 19.1% females (n = 1504). The majority of the sample was aged between 31–40 years (n = 2367; 30.1%) followed by the age group of 26–30 years (n = 1903; 24.2%). The age group with the least number of customers was 50+ years (n = 538; 6.8%), followed by the 18–20 age group (n = 564; 7.2%).

### 2.2. Gambling Website Description and Procedure

The authors were given access to a large anonymized dataset of customers at *Unibet* in order to carry out secondary analysis. This online gambling company offers a range of online products, including casino games, poker, sports betting, and in-play sports betting. The company also offers a range of responsible gambling tools as part of their commitment to player protection. One of these responsible gambling tools is VSE. The VSE option is something that the customers can do on their own and once this is done, the customer enters into an agreement that the account is suspended for the period that the customer has chosen. The VSE option is always available for the customer to use in their ‘Accounts’ section and information and a link to the tool is available on the operator’s dedicated RG page. In the case where a customer discloses that they have a self-reported gambling addiction, customer service agents at *Unibet* raise this issue with the responsible gambling department and the account is suspended immediately and permanently. The data collected for these customers were the gambling expenditure by the customer during their time with *Unibet*. The number of days leading to the VSE and closure due to self-reported gambling addiction was also obtained. Analysis of all data was carried out using *SPSS Version 27.* The data were all anonymized so that no customer profiles were identifiable to the researchers.

### 2.3. Data Analysis

Descriptive statistics were used to calculate means and percentages. In order to analyze the differences amongst the groups of customers, *t*-tests, one-way analysis of variance (ANOVA), and effect sizes were calculated using SPSS 23.0. This statistical analysis was carried out to examine the gambling behavior prior to VSE, and to compare gambling expenditure between those who utilized VSE and gambling expenditure among those with self-reported gambling addiction. The significance level for statistical analyses was *p* < 0.01.

## 3. Results

The initial descriptive analysis suggested that most customers that used VSE did not have significant gambling activity prior to self-excluding with *Unibet*. These customers might have had activity with other operators, but with this operator, these customers did not have significant gambling activity prior to self-excluding. When looking at customers who used VSE during this period, the majority of these customers (50.38%) actually self-excluded in the first seven days of activity. A further breakdown of the gambling activity by the time period prior to VSE can be seen in Table 1. 

### Differences in Gambling Expenditure by Days Leading to VSE Compared to the Addiction Group

The VSE groups that were split by the number of days of activity leading to the VSE in the five time periods (i.e., first day, first week, first month, first three months, and over a three-month period) were compared to the customers that closed their account due to self-reported addiction (Table 2 and Table 3; Figure 1). The total mean gambling expenditures were highest for the group that had self-reported gambling addiction, and the lowest was for the customers who used VSE in the first day. A one-way analysis of variance (ANOVA) was used to examine the effects of days leading to VSE and closure to self-reported addiction on gambling expenditure. There was a statistically significant difference between groups as determined by one-way ANOVA (F[5,7867] = 12.144, *p* < 0.0001). A Tukey post hoc test showed that gambling expenditure was significantly different with VSE in the first day (200.5 ± 546.3, *p* < 0.0001), VSE in the first week (305.6 ± 1041.3, *p* < 0.0001), VSE in the first month (362.2 ± 8062.8, *p* < 0.0001), VSE in the first three months (845.7 ± 3308.8, *p* < 0.0001), and VSE after three months (593.3 ± 2049.4, *p* < 0.0001), when compared to the customers that closed their account due to self-reported addiction (2584.4 ± 9223.4) with self-reported gambling addicts spending more money gambling than those who did not report gambling addiction across all time groups.

## 4. Discussion

The aim of the present study was to understand better whether VSE may be used as a reliable proxy measure for problem gambling. The main findings showed that the group of customers that use VSE is inherently different and not at all homogenous; therefore, it is rather limiting to consider it as one group. This varies greatly in terms of the days prior to using VSE, highlighting that it might be naïve to place all these gamblers under one umbrella. In fact, when looking at the time period of gambling activity prior to VSE, almost one-fifth of the sample had no gambling activity prior to self-excluding. Although these customers might have gambled elsewhere, on their *Unibet* account, they registered an account and self-excluded. A majority of the sample resorted to VSE within the first week of gambling activity (50.38%), which is similar to that reported in other studies [39,45]. What is quite evident is that although in previous studies customers who use VSE are regarded as one group [16,34,39,46], it is evident that the time period of gambling activity leading up to VSE varies significantly. Therefore, gambling operators who use customers who have voluntarily self-excluded as a proxy measure of problem gambling need to be cautious in using VSE as a potential marker of gambling harm. Customers that use VSE within a few days of registering the online gambling account should not necessarily be viewed as problem gamblers, and operators should monitor the more gambling-intense customers that have gambled on the website for at least a month. The key novelty of the present study is that it is the first (i) where the VSE group was split based on the amount of gambling activity prior to self-exclusion, and (ii) that customers using VSE were compared to customers that had used VSE but were also confirmed gambling addicts based on a self-report to customer services.

The VSE group split in accordance to different time periods (first day, first week, first month, etc.) was compared to the group of players with self-reported gambling addiction. The differences in the gambling expenditure of each group was significantly different, with the self-reported gambling addicts having the largest mean expenditure compared to the other groups analyzed. Statistically significant differences were found between the customers using VSE in the first day, first week, and first month compared to the customers using VSE in the first three months. It was also found that the group of customers who used VSE in the first three months had a higher mean gambling expenditure when compared to the group of customers who used VSE after the first three months. This was mainly due to a small number of very heavy spending gamblers, which increased the overall mean gambling expenditure. When looking at the mean differences between groups, it is worth noting that the largest mean differences were present when the groups were compared to the self-reported gambling addiction group. It is also worth noting that the mean gambling expenditure in the first group was lower than all the other means. It might also be concluded that the customers that chose to self-exclude with lesser gambling activity prior to using VSE may not have enough gambling activity to be considered in the same way as the customers that had a significant amount of gambling activity. Customers that experienced self-reported gambling addiction communicated with the gambling operator that they were experiencing gambling problems and/or gambling addiction. These accounts are closed as part of the company’s commitment to responsible gambling and harm minimization. It was expected that if VSE is a good proxy measure for problem gambling, then there would be similarities in the gambling expenditure with the group of customers that closed their account because of self-reported gambling addiction. However, no such similarities were found.

The findings of the research here are beneficial for operators, researchers, and policymakers because it provides insight into gambling behavior by analyzing real player behavior using tracking technologies, which is objective and unbiased [25]. Better understanding of the activity of online gamblers using transaction data arguably provides better data on which responsible gambling tools might be best used and offered to online gamblers. Given that online gambling websites offer the potential for innovation in responsible gambling tools [14], such studies are needed in order to actually understand the online gambling population.

When examining the findings as a whole, it appears that a large number of gamblers in the sample may have used VSE as a means to close their account for non-responsible gambling reasons, especially because these customers had little to no gambling activity prior to closing their account. Half of the customer population (50.38%) used VSE in the first seven days of opening their account. Based on this finding, it is arguably unreliable to use self-exclusion as a possible indicator or proxy measure for problem gambling. Online gambling operators and regulators are constantly looking into providing the best possible support for gamblers [44], but some online responsible gambling interventions appear to be replicas of what is offered in the land-based sector without proper evaluation of how or whether such measures would work online. The present study compared the group of customers that used VSE with a group of customers that confirmed self-reported gambling addiction to customer services. If VSE is a good proxy measure of problem gambling, then there should have been a close similarity concerning gambling expenditure with the group that confirmed gambling addiction, since this is based on self-reported problem gambling. However, this was not the case.

This lack of activity prior to VSE activation has also been reported in previous studies. For example, Dragicevic et al. [29] reported a high percentage of customers self-excluded within the first 15 days of gambling, including 25% of VSEs within the same day of account registration. The possible explanation given was that VSE might have been an impetuous decision. VSE online is arguably less shameful, stigmatic, and/or embarrassing than VSE in the land-based sector, and it can be done more easily and impulsively for non-responsible gambling reasons. In the study conducted by Hayer and Meyer [45], it was noted that most self-excluders considered the reason for VSE quite spontaneous and that a large proportion of the self-excluders did so for non-responsible gambling reasons (e.g., due to annoyance with the operator, as a preventive measure or at the request of third parties). Another possible reason might be due to the fact that the customer was approached via promotional email communication, and just wanted to remove access to the account that in turn might stop such promotions from being sent. Although there are probably customers who self-excluded for responsible gambling reasons, it might be that this proportion of the customer base self-excluded more as a rash decision or due to issues with the gaming operator. The reason for this is that when comparing the two VSE groups with each other, there are clear differences showing that those utilizing VSE are too heterogeneous to be treated as a single group. Moreover, when examining the main VSE group and the group that confirmed they had a gambling addiction, the differences were significant which further support the notion that VSE may not be the best proxy measure of problem gambling. However, the behavioral tracking data in the present study do not provide verification for these speculations. Therefore, further research directly asking about reasons for VSE is required.

Further research should examine the group of customers that initiate VSE without much gambling activity especially since it is such a significant proportion of the total of those who self-exclude within the first week of opening an account. Better understanding of the customer base, including self-excluders, would actually help the gambling industry in achieving profit without any exploitation of its customers. Research is also needed on self-excluders who have little gambling activity prior to self-excluding because these customers may be identified and advised in better ways to regulate their gambling, possibly by using other RG tools such as limit-setting. In this manner, the operator would be able to help the customer retain a sustainable relationship with the operator and not being potentially exploited by another operator. Future studies may also include other variables to be investigated such as the socioeconomic status of the customers, including the gambling status and more in-depth analysis on different aspects of the gambling behavior. Further studies may also examine the different types of gambling activity influence prior to VSE, the influence of time, and the impact of direct communication from the operator. Other future research could also include comparative analysis between customers using VSE and customers that do not. Moreover, the VSE group was split into unequal time intervals (i.e., one day, one week, one month, three months) in the dataset provided by the gambling operator. These are also typically the time intervals used by gambling operators for shorter-term VSE options (e.g., many operators provide gamblers with short ‘cooling off’ periods of one day, one week, or one month). These periods have nothing to do with problem gambling but are tools to help players gamble more responsibly. Future studies may also consider splitting the VSE group into time intervals that are more equally spread (e.g., every week or every month).

The present study is not without limitations. The data collected only comprised one online gambling operator and therefore it is unrepresentative as has been noted by others (e.g., [47]). Furthermore, due to the anonymity of internet gambling, the online gambling account can be shared with others and the customer may have more than one account [48] although the authors believe the number of gamblers that would be doing that in the present study would be very low. The participants in the present study might have had several online gambling accounts and therefore the activity evaluated may not have shown an accurate picture of the total gambling activity by the participant. A further limitation is that there was only one group comprising self-reported gambling addicts, whereas the VSE group was split into groups who had gambled for different durations. The reason for this was that the group comprising self-reported gambling addicts was too small to sub-divide any further. Future research would benefit from larger samples of gambling addicts so that they could be examined in terms of different lengths of gambling duration like those in the different VSE groups. Despite its limitations, the present study sheds further light on how customers’ behavior prior to VSE or prior to closing an account due to self-reported gambling addiction occurs. Through the better understanding of customer behavior, the gaming operator can engage in using different methods of communication, to the possible extent of helping customers refrain from losing too much of their disposable income, which may correspond to possible harm. Therefore, through this understanding and evaluation of such studies such as the present one, operators can proactively contribute to harm minimization. Harm minimization would have a direct impact on not only individuals suffering from gambling disorder, but also on their family and the communities.

## Figures and Tables

**Figure 1 ijerph-18-02000-f001:**
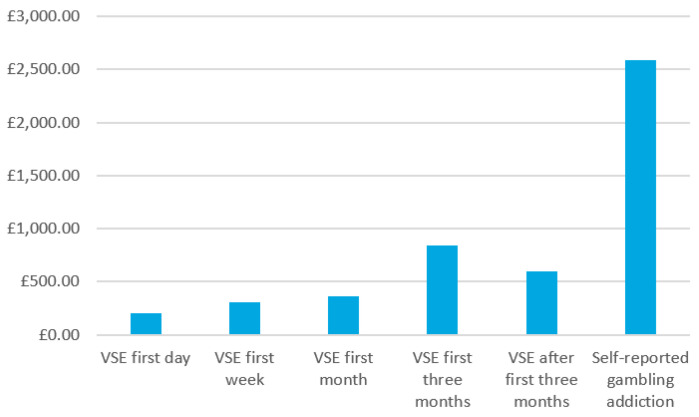
Total mean gambling expenditure in British pound by the VSE group and self-reported addiction closure group.

**Table 1 ijerph-18-02000-t001:** Number of days of gambling activity by gamblers prior to voluntary self-exclusion (n = 7732).

Time Period of Activity Prior to VSE	Percentage of Customers
0 days	19.15%
1–7 days	31.23%
8–30 days	17.85%
31–90 days	10.79%
91+ days	20.98%

**Table 2 ijerph-18-02000-t002:** Total mean gambling expenditure in British pound of the VSE groups and self-reported addiction account closure group.

Group	N	Mean	Standard Deviation
VSE first day	1481	£200.5	£546.15
VSE first week	2271	£305.6	£1041.12
VSE first month	1499	£362.2	£8060.10
VSE first three months	1274	£845.7	£3307.52
VSE after the first three months	1207	£593.3	£2048.55
Self-reported gambling addiction	141	£2584.4	£9303.20

**Table 3 ijerph-18-02000-t003:** The differences in total mean gambling expenditure in British pound of the VSE group and self-reported addiction closure group.

Group 1	Group 2	Mean Difference	*p*-Value	Effect Size
VSE first day	VSE first week	−£105.09	0.973	
	VSE first month	−£161.73	0.890	
	VSE first three months	−£645.20	<0.001 *	0.27
	VSE after the first three months	−£392.78	0.131	
	Self-reported gambling addiction	−£2383.93	<0.001 *	0.36
VSE first week	VSE first day	£105.09	0.973	
	VSE first month	−£56.64	0.998	
	VSE first three months	−£540.11	<0.002 *	0.01
	VSE after the first three months	−£287.69	0.357	
	Self-reported gambling addiction	−£2278.84	<0.001 *	0.34
VSE first month	VSE first day	£161.73	0.890	
	VSE first week	£56.64	0.998	
	VSE first three months	£483.47	0.024	
	VSE after the first three months	−£231.05	0.689	
	Self-reported gambling addiction	−£2222.20	<0.001 *	0.26
VSE first three months	VSE first day	£645.20	<0.001 *	0.27
	VSE first week	£540.11	<0.002 *	0.22
	VSE first month	£483.47	0.024	
	VSE after the first three months	£252.43	0.640	
	Self-reported gambling addiction	−£1783.72	<0.001 *	0.25
VSE after first three months	VSE first day	£392.78	0.131	
	VSE first week	£287.69	0.357	
	VSE first month	£231.05	0.689	
	VSE first three months	−£252.43	0.640	
	Self-reported gambling addiction	−£1991.15	<0.001 *	0.30
Self-reported gambling addiction	VSE first day	£2383.93	<0.001 *	0.36
	VSE first week	£2278.84	<0.001 *	0.34
	VSE first month	£2222,20	<0.001 *	0.26
	VSE first three months	£1738.72	<0.001 *	0.25
	VSE after the first three months	£1991.15	<0.001 *	0.30

* significant at the *p* < 0.01 level.

## Data Availability

The study contains commercially sensitive data, so access is restricted.

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
