# Peer review of "Understanding Online Voluntary Self-Exclusion in Gambling: An Empirical Study Using Account-Based Behavioral Tracking Data"

_ijerph, 2021, doi:10.3390/ijerph18042000_

Round 1
Reviewer 1 Report
Understanding online voluntary self-exclusion in gambling: An empirical study using account-based behavioral tracking data
There is growing consensus internationally that gambling can be a source of serious harm, disproportionally affecting vulnerable populations and recognised as a salient public health issue. Several studies have previously utilised voluntary self-exclusion (VSE) measures as a proxy indicator for problem gambling. The current study examined customers of a popular gambling operator (Unibet) who had engaged in VSE online. The findings highlight that VSE appears to be a poor indicator of problem gambling and that customers who used VSE are distinct overall, thus should not be grouped together homogeneously. Notable differences in engagement (i.e. time spent gambling) and expenditure were observed, particularly among those reporting gambling addiction. While this is a novel study incorporating objective gambling-related data (i.e. real player behavioural tracking technologies), there are a number of areas identified in the manuscript that require further clarification. I have noted my comments below and hope these are helpful to the authors.
Introduction
Overall, I suggest including a more comprehensive description of gambling-related harms, especially given the recognition of gambling as a growing public health issue.
Page 3: The last paragraph of the Introduction section would benefit particularly by incorporating further recognition of the potential benefits associated with providing tailored help to detect problem gambling i.e., extending beyond the individual and enhanced operator revenue (e.g. mitigating against adverse impacts social, economic etc on wider families and communities etc).
Materials and Methods
Page 3 (2.1 Participants and procedure): More data on the sample characteristics would be enlightening. For example, socioeconomic status and further details on gambling status, behaviour, or type etc. The information could be presented more concisely by incorporating a further table.
Page 3-4 (2.2 Gambling website description and procedure): Please elaborate further on how the data were shared between the gambling operator and the research team (anonymously, data sharing agreement etc). Further, please provide further details on the data management process. Also, were institutional ethical procedures required and if so, how were these adhered to or if not, please explain why?
Page 4: “The VSE option is something that the customers can do on their own and once this is done, the customer enters into an agreement that the account is suspended for the period that the customer has chosen”. Please outline when and how the gambling VSE options are explained and/or made available to the customers?
How were the data cleaned and who did the analysis (e.g. please describe quality assurance or checking procedures, distribution assessment etc.)? Also, more details on the precise statistical analysis employed would be beneficial to the reader, including the statistical significance level/alpha etc.
Results
Page 4: Table 1. I suggest explicitly including ‘gambling’ in the following section of the Table i.e., Time period of gambling activity prior to VSE.
Page 4: Table 2. "The gambling expenditure mean in GBP of the VSE groups (N=7732)". Please provide the full definition of GBP i.e. the first time described in the table text etc.
If further data on the type of gambling are available would it be possible to extend the analysis to examine if different types of gambling activity influence VSE engagement and/or expenditure (e.g., the type of gambling activity i.e., sports betting or online casino games)?
Discussion
Page 6: Paragraph two is a long summary of the Results with no integration of previous or other related literature (i.e., until paragraph three). I suggest bringing in further comparison of the current study findings with other studies and revise the second paragraph to make the Discussion flow more concisely.
Page 7: I suggest including some recognition that in future it would be enlightening to examine the potential influence of the type of gambling behaviour among customers who initiate VSE (or not) and how this may relate to gambling activity (time) as well as considering the amount and type of direct communications from gambling operators (including emails and text messaging) and whether opt out procedures influence VSE engagement.
Moreover, further details could be integrated into the final paragraph implying that greater understandings of consumer behaviour in future may be harnessed to potentially limit wider gambling-related harms (see earlier points on the Introduction) and encourage greater corporate responsibility among key gambling operators.
Reviewer 2 Report
The paper “Understanding online voluntary self-exclusion in gambling: An empirical study using account-based Behavioral tracking data” is relevant bacause utilized a big sample of persons self-excluded of one on-line gambling operator, and the study is one of the firsts to made a carefully analyse of this question. The design of the study is corect, the sanple big, and the statistical análisis adequate to the objective of the study. Many results were descriptive but very interesting in this area. The discussion is correct and include goods comments about the relation between self-exclussion and problem gambling. Is curious the data that many person begin to gamble and in few time made a self-exclussion. This is a interesting result of the study.
Reviewer 3 Report
The paper entitled “Understanding online voluntary self-exclusion in gambling: An empirical study using account-based behavioral tracking data” aims to evaluate different aspects of voluntary self-exclusion (VSE) as a proxy measure for problem gambling. The researchers carried out a secondary analysis of a large and anonymized dataset in order to assess the gambling expenditures, the number of days leading to the VSE and self-reported gambling addiction.
The research is well conducted, use account-based behavioral tracking data and adds empirical pieces of evidence to some debated question related to using VSE as a tool for responsible gambling. Moreover, it stimulated some questions that I’d like the authors should address.
The authors provide convincing evidence regarding the heterogeneity of the sample of VSE customers and highlight the prevalence of people who self-exclude in the first seven days. Moreover, they interpret this detail as a confirmation that VSE is not a proxy of problem gambling but is related to other reasons as a rash decision or issues with the gaming operator. Both hypotheses necessitate of a direct verification that the tracking data analyzed cannot support. To further characterize these customers, perhaps the authors should have compared them with other gamblers with the same duration of registration of their account, who have not used the VSE. This comparison could perhaps dispel the suspicion that the observed increase in GPD in table 2 can be due to the cumulative effect of time elapsed since account registration. Moreover, the comparison of the two trends could clarify a curious result that interrupts the linear slope: the gambling expenditure of the group which led to the VSE after three months from the registration of their account. This result was not discussed but I think it deserves some attention (see fig. 1). Finally, the comparison of gambling expenditure mean (GPB) of VSE groups and self-reported addiction closure-group, as was performed, presents some redundancy (see the first five-row of tables 2 and 3) that a simple comparison between the means of two independent groups could solve.
Reviewer 4 Report
Thank you for the opportunity to review this paper. The study used a one-way ANOVA to examine differences in gambling expenditure between six groups of individuals who used voluntary self-exclusion (VSE): (1) after the first day, (2) first week, (3) first month, (4) first three months, (5) more than three months, and (6) self-reported addiction. The authors concluded that VSE was not a good proxy measure of gambling problems because there was a significant difference between mean expenditure amounts across the six groups. While this study examines an interesting topic, there are numerous issues that require the authors’ attention:
In the introduction, the authors mention “gambling-related harm” but it is not clear what this is. Some discussion of possible harmful consequences seems appropriate.
The introduction discusses the lack of regulation in the beginning but then indicates that it’s in the interest of operators to invest in consumer protection because irresponsible behavior can result in fines from regulatory bodies. How much regulation is there? Further discussion of this seems to be needed.
The authors claim that in a study conducted by LaPlante et al., “episodic increase in betting activity... is almost certainly due to the availability of sports events.” Is this a conclusion that LaPlante et al. made based on evidence or is this an assumption?
Rather than cite a primary source (e.g., Hayer & Mayer), the authors self-cite a commentary piece (i.e., Griffith & Auer) when suggesting VSE is a bad proxy measure. This is not recommended practice.
Discuss the number of participants and whether it met your priori sample size determination. Also, were any participants excluded from the data set? If so, why?
The authors simply report that there is a significant difference found among the groups. Were all groups significantly different from each other, or just some? If it’s only some, which ones are they? This additional level of analysis is needed.
A rationale was not provided for the inclusion of gambling expenditure as a dependent variable. This should have been discussed in the introduction section.
I did not see any effect size provided in addition to the results of significance tests.
The authors conclude that VSE is just not a good proxy measure for problem gambling. But is this true for all groups? Is it as true for those that used it after one day of gambling as it is for those who used it for three months?
The authors cited studies that used several different measures for problem gambling. How does VSE compare with these other measures?
Overall, the conclusion does not seem to be strongly supported by the results of the analyses.
Many statements are awkwardly worded. There are spelling and grammatical errors, and the language used throughout the paper can be further improved.
Round 2
Reviewer 4 Report
The authors have made some improvements in the resubmitted version of the manuscript. However, there continues to be issues that prevent me from recommending publication. The following is a list of my major (and a couple minor) concerns.
The Griffiths & Auer article is a brief commentary piece that cited research conducted and summarized conclusions made by others. Primary sources are authoritative pieces that offer a firsthand account of the subject of interest. The Griffiths & Auer is not a primary source.
Also, the line graph is not appropriate as a visual display of mean gambling expenditures across groups. Line graphs show change over time and would be okay if the authors were looking at the same subjects across time, but these are different groups. Use a bar graph.
The mean expenditures reported after the first 3 months is less for the group that self-excluded after 3 months compared to the group that self-excluded within the first 3 months. What are possible reasons for this?
In looking at the groups created, the time intervals that define them are highly unequal. For example, the difference between the first two groups (VSE first day and first week) is a period of 6 days, but the difference between the fourth and fifth groups (VSE for first month and first three months) is much bigger. How were these ranges determined? And how might such groupings result in limitations in the interpretation of the results?
While a table was added to report mean differences between groups, there was no interpretation of the results presented nor discussion of what they mean. This seems necessary given the varying differences between groups.
Is there any information about the length of time that self-reported gamblers gambled? It seems to me that this would be an important variable that may influence interpretation of results given the potential difference in opportunity/time to gamble between the groups.
